# Optimization of Sensory Properties of Cold Brew Coffee Produced by Reduced Pressure Cycles and Its Physicochemical Characteristics

**DOI:** 10.3390/molecules27092971

**Published:** 2022-05-06

**Authors:** Smaro Kyroglou, Rafailia Laskari, Patroklos Vareltzis

**Affiliations:** Chemical Engineering Department, Laboratory of Food and Agricultural Industries Technologies, Aristotle University of Thessaloniki, 54124 Thessaloniki, Greece; kyrosmar@cheng.auth.gr (S.K.); rflaskari21@gmail.com (R.L.)

**Keywords:** cold brew, coffee, vacuum, sensory evaluation, principal component analysis

## Abstract

The use of vacuum cycles for the cold extraction of coffee is a new process that leads to a significant reduction in process time of Cold Brew compared to conventional methods. This research aimed at specifying the necessary parameters for producing a consumer-accepted cold brew coffee by applying vacuum cycles. This was achieved by investigating the effect of the number of cycles and of the applied pressure (vacuum) on the physicochemical characteristics of the cold brew coffee, i.e., total dissolved solids (TDS%), pH, acidity, phenol and caffeine content and color. Furthermore, sensory evaluation took place by members of the Specialty Coffee Association of America (SCAA) to specify parameters such as coffee blend, coffee/water ratio, total water hardness and grind size and secondly to determine the optimal pressure and number of cycles for a tasty final beverage. The sensory and physiochemical characteristics of cold extraction coffee were investigated by Principal Component Analysis (PCA). It became evident that coffee extraction by applying two vacuum cycles at 205 mbar pressure produced the lowest intensity of physiochemical properties (caffeine, phenols, acidity, TDS% and pH), and the highest score of sensory characteristics (fragrance, body, acidity, flavor, balance, and aftertaste). Caffeine and phenol concentration of the optimal beverage were 26.66 ± 1.56 mg/g coffee and 23.36 ± 0.79 mg gallic acid/g coffee respectively. The physiochemical characteristics were also compared to a beverage of hot extraction of the same blend and ratio of coffee to water.

## 1. Introduction

Coffee is one of the most popular beverages in the world due to its pleasant aroma and taste. According to the International Coffee Organization, more than ten million t were consumed worldwide in 2021 [1]. Its consumption has been established in the daily routine of people because of the stimulating effects of caffeine and the high antioxidant activity [2,3,4], which is attributed to certain components of coffee beans, such as phenols, chlorogenic acids and melanoidins [5].

Hot brewing methods are mainly used for the production of coffee beverages and they vary depending on geographic, cultural, and social context, as well as on personal preferences [6,7]. However, a new method of brewing, called cold brewing, has gained popularity in recent years. It is a low or room temperature extraction process that needs a considerable amount of time for completion, ranging from 8 to 24 h [8]. Recent studies conducted by our team have shown that the duration of extraction is significantly shortened when it is performed under cycles of reduced pressure (vacuum cycles). This process led to short extraction time (65 min) and high caffeine concentration [9]. Also, previous research showed that the combination of ultrasonication and agitation by a stirrer led to increased soluble solids, higher antioxidant activity and a maximum caffeine concentration of 14.97 g/kg in comparison with the traditional method of cold brewing [10]. The ultrasound assisted extraction was also proposed by Zhai et al. (2022)., since it resulted in acceleration of extraction time, improvement of extraction efficiency and comparable sensory properties to original cold brew coffee [11]. Other researchers suggested that the cold brew extraction process can be accelerated by microwave heat treatment, but an initial heating step up to 80 °C is required [12]. Moreover, the flavor profile of cold brew coffee diverges significantly from the beverage prepared by hot extraction due to the longer extraction time. Specifically, it has been characterized as sweet and smooth, because the components with sweet flavor are soluble even in cold water, in contrast to oils and acids that are non-soluble [9].

Numerous studies have been conducted for the extraction process of coffee and the main factors affecting it. However, the research about the cold extraction, the physicochemical properties and the sensory profile of the final beverage is limited. Cordoba et al. (2019) noted that the main factors affecting the physicochemical characteristics of cold brew coffee were the degree of grinding and the extraction time [2]. Other researchers suggested that agitation had a significant effect in the physicochemical properties, where a significant increment in caffeine and phenolic content and antioxidant activity was mentioned [9,10]. Agitation significantly affected the physicochemical characteristics due to tissue and cell wall disruption and facilitated the transfer of soluble solids and caffeine through the cell membrane to the solution [10]. Moreover, Rao and Fuller (2018) showed that the pH values of cold brew coffees were higher than those produced by hot extraction, whereas the titratable acidity was found higher in hot brewing beverages [13]. Another study demonstrated that cold brew processes, similar to those for hot brew coffee, impart different flavors and flavor intensities in coffee, which ultimately led to different consumer liking scores [14].

Therefore, the objective of this study was to design a sensorially optimized cold brew coffee by applying the innovative vacuum cycle processing technique. For the first time, a correlation between the physicochemical and operational parameters of this process with the sensorial attributes of the coffee extract is reported. Specifically, the optimization was performed by investigating parameters, such as coffee blend, degree of grinding, coffee to water ratio and total water hardness, as well as the pressure and number of vacuum cycles. The produced beverages were sampled and tested by experienced SCAA members and results were analyzed by Principal Component Analysis.

## 2. Results and Discussion

### 2.1. Specification of Extraction Parameters: Blend, Coffee/Water Ratio, Total Water Hardness and Grind Size

#### 2.1.1. Coffee Blend

The coffee blend used for all experiments was Columbia–Tres Lomas Estate SHG. This choice was based on the opinion of the SCAA members fulfilling two main requirements: the blend should have strong aroma and fruity taste in order to maximize the aromatic and sensory characteristics that could be transferred from the ground coffee to the extract, when vacuum cycles are applied and secondly, the coffee blend should be a relatively common commodity in the local coffee market.

#### 2.1.2. Number of Vacuum Cycles for Optimizing Extraction Parameters

Before optimizing the rest of extraction parameters, i.e., coffee/water ratio, total water hardness and grind size, it was necessary to choose an appropriate number of vacuum cycles for these experiments. Starting point of the experiments was the previously published results of 13 cycles, 205 mbar, 1/9 coffee to water ratio and total water hardness 12 ppm, which maximize solids yield [9].

Preliminary sampling tests by SCAA members showed that 7 vacuum cycles resulted in a sensorially accepted beverage. Furthermore, caffeine and phenolic content of coffee produced by 7 vacuum cycles did not differ significantly compared to that of the extract produced by applying 13 vacuum cycles (Appendix A). These data, in conjunction with the fact that the extraction process could be completed in a shorter period of time by applying 7 vacuum cycles instead of 13, led to test both of these values. Therefore, two treatments were examined. Specifically, in treatment A, 13 vacuum cycles were applied at coffee/water ratio 1/9, whereas in treatment B the vacuum cycles were 7 at the same ratio (Table 1, First test).

SCAA specialists tested and compared beverages from treatments A and B. The coffee samples produced by the first treatment were characterized as “over-extracted” and without taste and flavor. On the other hand, the coffee produced by the application of 7 vacuum cycles at ratio 1/9 (treatment B) was balanced and tasteful. Consequently, 7 vacuum cycles were selected as the optimum value under these conditions of extraction. The characterization “over-extracted” coincides with the fact that previous research has shown that cold extraction, under similar to treatment A conditions, led to maximum solids yield [9]. However, these solids, including caffeine, phenols, chlorogenic acids, esters and organic acids, should contribute to the taste and flavor of the extracted coffee. It is suggested that the large number of vacuum cycles might contribute to the removal of aroma and flavor compounds, especially the volatile ones, rendering the over-extracted sample tasteless.

#### 2.1.3. Total Water Hardness and Coffee/Water Ratio

In the second set of experiments (Table 1, Second test), the 12 ppm total hardness water, was found to be inappropriate for cold coffee extraction. According to the specialists it resulted in over-extraction and bitter taste compared to the extracts with higher total water hardness. Similar conclusions were reached by Hendon et al. (2014), who suggested that calcium and magnesium are major contributors to the efficiency of extracting flavorsome coffee compounds. Specifically, the dissolution and extraction of the organic molecules contained in coffee beans is a process dependent on the dissolved mineral content of the water [15]. Consequently, the absence of minerals that characterizes distilled water resulted in unacceptable extraction levels. On the other hand, the water of total hardness 70, 109 and 290 ppm led to acceptable extraction levels. According to the SCAA an upper limit of dissolved solids in water (ca. 300 ppm TDS) is suggested for favorable extraction of coffee [16].

The optimum combination of water hardness and ratio was determined using sensory analysis and the Coffee Brewing Control Chart, presented in Figure 1 [17]. The examined ratios 1/14, 1/16 and 1/18, all calculated as w/w, are depicted in this chart. It should be noted that this chart is used mainly for characterizing hot extraction of coffee (filter coffee), where the suggested ratios range from 1/12 to 1/20. Therefore, the ratio 1/9 is not included in the chart. Cold brewing is a new extraction method and the chart for this specific process has not been designed yet. Also, as it can be seen in Figure 1, the ratio 1/9 could be assumed to lie in the upper left corner of the chart and so, the three points that correspond to the examined hardness (70, 109 and 290 ppm) are located in the region of strong and under-extracted.

Moreover, the measured TDS% of the extracts produced by coffee/water ratio 1/14, 1/16 and 1/18 and water hardness 70, 109 and 290 ppm ranged from 1.50 to 1.60. Each sample was tested by the specialists panel and was given a score, which is depicted in Figure 1. The points that correspond to ratio 1/16 and 1/18 are located in the region of strong and over-extracted. Therefore, they were deemed inappropriate for favorable extraction under these conditions. On the contrary, the ratio 1/14 was the optimum value, because the points were found in the colored region characterized as strong but close enough to the region of “SCA ideal”. Particularly, the point that corresponds to water hardness 109 ppm was the closest to ideal and this was confirmed by the specialists who found the coffee extract tasty and pleasant. This ratio was in agreement with the suggested one found in the literature for the production of a typical ready-to-drink cold brew coffee [18]. Consequently, the sensory profile of the beverage was enhanced by performing extraction with water hardness 109 ppm and coffee/water ratio 1/14.

#### 2.1.4. Selection of Coffee Grinding Degree

From the four grinding sizes examined in the third set of experiments (Table 1, Third test), the 800–1000 and >1000 μm particle size stood out. Specifically, the coarse grinding of 800–1000 μm resulted in a beverage rich in taste and aroma. Furthermore, the sample with grinding size >1000 μm was characterized by the specialists panel as very tasty and aromatic with interesting acidities, as well as a pleasant oily texture. The amount of lipids determined for each grinding degree are presented in Appendix A (Appendix A). Similar conclusions were reached by Cordoba et al. (2019), where the best scores for cold brewed coffees in terms of overall impact were for those samples brewed using coarse grinding. Apart from the sensory attributes, the coarse grinding led to increased total dissolved solids and phenolic content [2]. Finally, according to the specialists, the sensory profile of the extract produced by >1000 μm grinding size was preferable to the 800–1000 μm one.

Summarizing the above results, the optimum extraction parameters for a tasty cold brew coffee produced by vacuum extraction method are shown in Table 2. These parameters were kept constant for the rest of the experiments, where optimization of the operating parameters took place.

### 2.2. Optimization of Operating Parameters for the Vaccum Cycle Assisted Extraction

#### 2.2.1. Effect of Vacuum Cycles on Physicochemical Properties

The effect of varying the number of vacuum cycles (300 s each) at constant pressure (205 mbar) on the physicochemical properties of the cold extracted coffee are presented in Table 3. Vacuum cycles had a significant effect on the caffeine content, phenols, acidity, pH, TDS% and ΔΕ (*p* < 0.05). Specifically, the extracted caffeine increased by increasing the number of vacuum cycles. Maximum caffeine concentration was achieved at 10 cycles and did not significantly increase after 13 cycles. Similar trend was noted in the TDS% and the extracted phenols, where the highest value was achieved after 10 vacuum cycles. This can be explained by the fact that the physical barrier that water meets when trying to enter the coffee beans degrades with each new vacuum cycle and as a result, the extraction process is facilitated. The enhancement of mass transfer was also mentioned by Wang et al. (2014), who found that vacuum assisted extraction, applied to extract bioactive components from *Andrographis paniculata*, resulted in shorter extraction times and remarkable higher extraction efficiency than ultrasonic assisted extraction and heat reflux extraction. These results indicated that a certain degree of vacuum allowed a better penetration of the solvent into the pores and between the matrix particles [19].

As the number of vacuum cycles increases, the concentrations of extracted components, such as caffeine and phenols, also increase (Table 3). However, after a certain number of vacuum cycles, most of the extractable components have been transferred into the extract and therefore, their concentration is stabilized and/or decreased. The decrease of phenols after applying 13 and 16 vacuum cycles can be attributed to the destruction of the cell structure of ground coffee because of the prolonged vacuum cycles applied. This treatment leads to the release of oxidative and hydrolytic enzymes and thus to the destruction of phenols. Similar results were obtained for oyster mushrooms and tomatoes that were microwave dried [20,21].

It has been reported that extraction under reduced pressure achieved maximum caffeine and phenolic content at 13 vacuum cycles [9], while in this work maximum levels of these components were achieved at 10 cycles. This difference may be attributed to the different coffee blend used for the experiments, as well as to the difference in total hardness of water used for the extraction. The presence of dissolved ions (Mg^2+^ and Ca^2+^) in the water facilitated the extraction process, resulting in increased caffeine and phenolic content by applying fewer vacuum cycles [15]. Furthermore, the correlation of caffeine and TDS% with color ΔΕ was found statistically significant up to 10 vacuum cycles. The Pearson correlation coefficient of caffeine and TDS% with ΔΕ were −0.735 and −0.819 respectively. This means that the variation of caffeine and TDS% caused by applying more vacuum cycles significantly affected the color of the extracted coffee.

#### 2.2.2. Effect of Pressure on Physicochemical Properties (2 and 7 Vacuum Cycles)

Sensory evaluation (Figure 2 and Figure 3) revealed that 2 and 7 vacuum cycles were the most preferable for producing a tasteful extract. Therefore, we investigated the effect of different pressure values at 2 and 7 vacuum cycles. In Appendix A summarize the statistical analyses of Figure 2 and Figure 3.

The results of physicochemical properties as affected by pressure for 2 vacuum cycles are presented in Table 4. Pressure didn’t have a significant effect on the caffeine content and TDS% (*p* > 0.05). On the other hand, the phenolic content, acidity, pH and ΔΕ were affected significantly (*p* < 0.05) by pressure for 2 cycles. Specifically, the extraction of phenols was enhanced at low pressure conditions. At these conditions, the high vacuum facilitates the cell wall disruption of coffee beans and the extraction process of phenolic compounds. Similar results were found by Ranjbar et al. (2016), Xie et al. (2015) and Wu et al. (2019), who noticed that the available active ingredients in plant materials were higher and purer when extracted under reduced pressure conditions [22,23,24]. This may be due to the fact that the operation in a relatively low temperature environment avoids the degradation and oxidation of heat-sensitive and easily oxidized components [24]. Furthermore, there was no significant correlation of caffeine content and TDS% with the parameter ΔΕ.

The results of physicochemical properties as affected by pressure for 7 vacuum cycles are presented in Table 5. Statistical analysis showed that pressure didn’t have a significant effect on the caffeine content, TDS% and ΔΕ (*p* > 0.05). On the other hand, the phenolic content, acidity and pH were affected significantly (*p* < 0.05) by pressure for 7 cycles. Moreover, the correlation of caffeine and TDS% with ΔΕ was not statistically significant.

### 2.3. Descriptive Sensory Evaluation

The scores from the sensory test were used for the design of spider diagrams. According to Figure 2 the beverage which was distinguished for the intense sensory profile was produced using 2 vacuum cycles, whereas 16 vacuum cycles produced a beverage with the lowest intensity of sensory properties.

Comparing the physicochemical properties of these beverages, it is concluded that caffeine and titratable acidity were significantly higher in the sample of 16 cycles (Table 3). However, the acidity determined by sensory analysis was higher in 2 cycles (Figure 2). This result comes in agreement with other researchers who reported no correlation between titratable acidity and perceived acidity [7,25]. This difference may be explained by the fact that changes in coffee brew acidity are slight, and the most common methods to measure acidity (alkali titration) cannot detect small changes [7,26]. In addition, there could be an overlap of perceived acidity with other sensory attributes [26]. Furthermore, it was observed that the coffee brew of 2 cycles had the strongest aroma, while most of the sensory properties’ intensities degraded by increasing the number of vacuum cycles. Moreover, statistical analysis showed that vacuum cycles had a significant effect on all the organoleptic characteristics evaluated during the sensory test. Specifically, aroma, flavor, acidity, aftertaste, body and balance were affected significantly by the vacuum cycles (*p* < 0.05).

In Figure 3a the spider diagram for 2 vacuum cycles and variation of pressure is presented. It is noticed that sensory attributes do not follow a linear relationship to increased pressure. For example, increasing the pressure from 20 to 100 mbar, acidity is also increased, whereas higher pressures lead to decreased acidity. This can be attributed to the phenolic content of the coffee extracts. Farah et al. (2006) supported the idea that coffee’s sour and astringency flavor is mainly provided by phenols and their degradation products [27]. According to Table 4, the phenols extracted for pressure ranging from 20 to 100 mbar are significantly higher than those extracted for pressure ranging from 300 to 700 mbar. Hence, the decreased acidity observed for higher pressures is due to the low phenolic content under these conditions.

Furthermore, the highest intensity of flavor is observed for pressure of 100 mbar. The high phenolic content determined after extraction process at 100 mbar is responsible for the intense flavor, as it was mentioned by Clifford [28]. It was also suggested that acidity and bitterness are the main sensory properties that were promoted by the intense flavor [28,29]. Our study confirms these previous researchers, because at pressure 100 mbar the highest flavor intensity is accompanied by the highest acidity.

Also, in Figure 3b, the spider diagram for 7 vacuum cycles and variation of pressure is presented. It is concluded that high values of pressure ranging from 300 to 700 mbar are not the optimal choice of the testers, because of the mild sensory profile, as it can be seen in Figure 3b. On the contrary, pressures of 50 and 100 mbar result in the most tasteful and preferable coffee extract. Specifically, all the sensory attributes received the best grading by the specialists for these two extraction conditions. Moreover, the high acidity determined by sensory evaluation for 50 and 100 mbar can be attributed to the significantly high content of extracted phenols in these pressures [28]. Finally, statistical analysis showed that all the sensory attributes mentioned above are affected significantly by pressure for 2, as well as, for 7 vacuum cycles.

### 2.4. Principal Component Analysis (PCA)

Principal Component Analysis was used in order to examine the correlation between physicochemical and sensory properties in the two series of experiments (test of vacuum cycles and pressure). In PCA physicochemical and sensory features with greater “power” are displayed on a two-dimensional surface, described by orthogonal factors used as dimensions (PC1 and PC2) to highlight differences or similarities between the analyzed samples [30].

#### 2.4.1. Test of Vacuum Cycles at Constant Pressure

The score and loading plot of the first series of experiments are presented in Figure 4a and 4b respectively. Two principal components represented more than 83.1% of the total variance in the physicochemical and sensory properties, and PC1 and PC2 of the PCA analysis model explained 58.4 and 24.7% of the variance, respectively. Moreover, all samples of five and seven vacuum cycles were found close together and located in the first quadrant. They were characterized by the highest intensity of the aftertaste, balance, flavor, pH and TDS%, whereas the intensity of aroma was very low. In the second quadrant, all samples of two cycles were identified, which showed the lowest intensity of all physicochemical properties (caffeine, phenols, acidity, TDS% and pH) and higher value of sensory attributes (aroma, body, acidity, flavor, balance, and aftertaste). The different trend that TDS% and body followed was not expected, because body during cupping is a proxy for the dissolved coffee solids (e.g., organic acids and oils, proteins, fibers) [31]. At this point it is worth mentioning that samples of two vacuum cycles at a constant pressure of 205 mbar were characterized by the trained panel, as excellent, tastefully comparable to commercial Cold Brew, with intense fruity aroma and low acidity. This intense aroma could be caused by the lipids in coffee, that form emulsions. Toci et al. (2013) reported that the emulsions retained aromatic compounds and as a result the aroma was strengthened, and the coffee beverage possessed a mellow and long aftertaste [32]. This observation is in accordance with the strong aftertaste determined by sensory testing for 2 vacuum cycles. Finally, in the third and fourth quadrant the samples of ten, thirteen and sixteen vacuum cycles were located, which were characterized by the highest intensity of caffeine and acidity and the lowest intensity of sensory properties.

#### 2.4.2. Test of Pressure

##### Two Vacuum Cycles

The score and loading plot of the second series of experiments are presented in Figure 5a,b respectively. Two principal components represented more than 66.1% of the total variance in the physicochemical and sensory properties, and PC1 and PC2 of the PCA analysis model explained 42.7 and 23.4% of the variance, respectively. Furthermore, all samples of 20, 50 and 700 mbar by applying two vacuum cycles were found close to each other and were identified in the third and fourth quadrant. They were characterized by the highest intensity of caffeine and phenols and the lowest intensity of all sensory attributes (aroma, body, acidity, flavor, balance, and aftertaste). The weak sensory profile produced by applying vacuum cycles at 20 and 50 mbar can be explained by the depression of boiling point under vacuum conditions. Specifically, the boiling point was calculated equal to 18 °C and 34 °C for pressure 20 and 50 mbar respectively [33]. Therefore, during the extraction process boiling was observed, which could have led to the loss of many volatile compounds and aroma of the extract. It can be concluded that coffee extraction at low or high values of pressure for 2 vacuum cycles is not the optimal choice because of the absence of intense sensory properties rendering the coffee beverage unacceptable for consumers.

Moreover, in the second quadrant, all 100 mbar samples were identified, which showed the highest value of aroma, flavor, balance, and acidity. While, in the first quadrant, the samples of 300 and 500 mbar were located and characterized by the highest intensity of acidity, aftertaste, TDS%, body and the lowest levels of caffeine and phenols. The TDS% and body followed the same trend, as it was expected, in contrast to the Section 2.4.1, where these properties did not correlate. According to these observations, the optimal pressure to produce a sensorially optimized cold brew coffee lies within the range of 100 and 500 mbar, where the sensory properties are of high intensity.

### 2.5. Comparison between Hot and Cold Coffee Extraction

The coffee extracts used for the comparison between hot and cold brewing were prepared under the same conditions (coffee blend, degree of grinding, coffee/water ratio and total water hardness). Furthermore, a comparison was made between the samples of hot and cold brewing and a sample of commercial cold brew coffee. The results of the physicochemical properties of hot and cold brew coffee (commercial and not) are presented in Table 6.

According to Table 6 phenols were significantly higher in hot coffee extraction, whereas caffeine and pH were significantly higher in commercial cold brew coffee. Generally, it is considered that caffeine content in hot brew is significantly higher than in cold brew for the same coffee to water ratio because high temperature facilitates the solubility of caffeine [9]. However, in this study cold brewing enhanced the caffeine content as it was observed also by other researchers, who suggested that the higher caffeine content in cold infusions may be caused by extending the brewing time, which increases the intragranular diffusion [8,34]. Also, other authors found that caffeine is comparable between hot and cold brewing [9,35]. Similar conclusions about pH were reached by Cordoba et al. (2019) and Rao and Fuller (2018), where cold brew coffees exhibited higher pH values (less acidic) than their hot counterparts [2,13]. They also found that total phenolic content was higher in hot brew coffees [2,11,36]. Farah et al. (2006) noticed that low total phenolic content makes cold brew coffee slightly sweet and tasty, because coffee’s sour and astringency flavor is mainly provided by phenols and their degradation products [11,27]. Other researchers suggested that cold brewing infusions of unroasted beans lead to higher content of these compounds, while the hot brewing technique seems to be more effective for roasted beans [34].

Moreover, acidity and TDS% of the measured coffee samples were not affected significantly by the extraction method. No significant differences were also found in TDS% by Cordoba et al. (2021) [37]. Moreover, recent studies reported that titratable acidity in hot coffee extracts was higher compared to cold brew coffee extracts, which could indicate that hot brewing was able to extract more acids and additional acidic compounds [2,13,37,38,39]. This difference could be attributed to the extraction method under reduced pressure used for the production of cold brew coffee, where the extraction time was significantly lower than that of traditional cold brewing and so the components transferred from coffee beans to the solvent may differ.

## 3. Materials and Methods

### 3.1. Raw Materials and Reagents

Medium roasted coffee beans of 100% Arabica variety (Columbia-Tres Lomas Estate SHG) were purchased from a local vendor (Pianeta Gusto, Thessaloniki, Greece). The distilled water of total hardness 12 ppm was produced by a water purification system (Hydrolab, R10, Straszyn, Poland), while the water of total hardness 70, 109 and 290 ppm was purchased from the local market. All chemicals and reagents were of analytical grade and obtained from Merck & Co. (Kenilworth, NJ, USA) and Laboratory of the Government Chemist (LGC) standards (Teddington, UK).

#### Experimental Procedure

Regardless the coffee extraction process, the main parameters that affect the sensory profile of the final beverage are grind size, coffee blend, coffee to water ratio and water total hardness [26]. Depending on the extraction technique, other parameters may play a role, such as temperature (e.g., hot extraction), pressure (espresso) [26,40,41]. In the case of vacuum pressure cycle method, the number of cycles and the value of pressure can affect the yield, as well as the sensory profile of the final beverage [9]. Therefore, initially, grind size, coffee blend, coffee to water ratio and water total hardness were optimized with the help of the SCAA members following the procedures described in Table 1 (Section 3.3.1). These optimized values were kept constant in the second set of experiments designed to optimize the vacuum cycles and pressure in regard to the sensory profile of the final beverage (Section 3.3.2). Figure 6 summarizes the experimental process and the variables to be optimized.

### 3.2. Sample Preparation

Coffee beans of the chosen blend were ground in different sizes with a coffee grinder on the day of the experiments. The grind size was determined using sieves with different mesh sizes ranging from 600 to >1000 μm.

The samples used for the experiments were prepared at different coffee to water ratios ranging from 1/9 to 1/18 (*w*/*v*). The required quantity of ground coffee was weighed with an electronic scale and placed in a plastic container, which contained the equivalent quantity of water (of specific hardness) at room temperature (20 °C).

### 3.3. Extraction by Applying Vacuum Cycles

The sample was covered with cling film in order to prevent spillage and placed in a vacuum pack machine (MULTIVAC, C200, Wolfertschwenden, Germany). Pressure and duration of each vacuum cycle were set for each experiment. At the end of the process, the coffee extract was collected by filtration [9].

#### 3.3.1. Specification of Extraction Parameters: Coffee/Water Ratio, Total Water Hardness and Grind Size

This part of experiments aimed at establishing optimized values of the extraction parameters and is presented in Table 1. Starting point of the experiments was the results published in a previous study: 13 cycles, 205 mbar, 1/9 coffee to water ratio and total hardness 12 ppm. Pressure and number of cycles were optimized for maximum solids yield [9]. For the determination of the extraction parameters Sensory Analysis was conducted by experienced members of SCAA.

The first test (Table 1) was performed in order to determine a suitable number of vacuum cycles to be applied for extraction.

Afterwards, the result of the 1st test was used to determine the total water hardness and coffee to water ratio (2nd test). The chosen values of hardness that were examined were 12, 70, 109 and 290 ppm. Although specialists suggested that water hardness of 12 ppm should not be examined due to the degraded sensory properties of the final extract, it was decided to be tested as a blank at ratio 1/9. The coffee/water ratios examined were: 1/9, 1/14, 1/16 and 1/18. The optimum combination of water hardness and ratio was determined using sensory analysis and the Coffee Control Brewing Chart, a useful tool for evaluating the final brew and the extraction level [17]. In order to use this chart, the coffee to water ratio and the total dissolved solids of the coffee extract should be known.

Finally, based on the results of the optimum hardness and ratio, the 3rd test was designed in order to select the grind size that produced an acceptable sensory profile of the cold brew coffee. The examined grind sizes were 600–700, 700–800, 800–1000 and >1000 μm.

#### 3.3.2. Optimization of Operating Parameters for the Vacuum-Cycle Assisted Extraction

##### Effect of Vacuum Cycles on Physicochemical Properties

The effect of vacuum cycles on the extract’s physicochemical and sensory properties was investigated by applying 2, 5, 7, 10, 13 and 16 cycles at a constant pressure of 205 mbar. The coffee/water ratio used was 1⁄14, the total water hardness 109 ppm and the grind size over 1000 μm as a result of the investigation carried out in the Section 3.3.1. The pressure for the experiments was set to be 205 mbar, which was found to render the maximum efficiency of caffeine and phenol extraction in a previous study [9].

##### Effect of Pressure on Physicochemical Properties at Constant Number of Cycles

The effect of pressure on the extract’s physicochemical and sensory properties was investigated by applying 20, 50, 100, 300, 500 and 700 mbar with 2 or 7 cycles (300 s each). This choice was based on preliminary experiments, where it was established that extraction by applying 2 and 7 vacuum cycles could produce a tasteful cup of coffee according to the specialists.

After the completion of the cycles, the extract was passed through the filter and collected in order to determine the physicochemical properties, i.e., total Dissolved Solids (TDS%), pH, acidity, phenol and caffeine content and color.

### 3.4. Physicochemical Characterization of Samples

#### 3.4.1. Percentage of Total Dissolved Solids TDS%

Total soluble solids were measured using an electronic refractometer (ATAGO, PAL-1, Tokyo, Japan) and were initially expressed as Brix%. After the measurements that were taken in Brix%, a conversion to TDS% was performed with the following equation proposed by Gómez [42]:(1)TDS%=0.85×Brix%

#### 3.4.2. pH

The pH of each brewed coffee sample was measured with a pH meter (HANNA, pH 211, Woonsocket, RI, USA).

#### 3.4.3. Acidity

The total titratable acidity of the coffee extracts was expressed in ml of 0.10 N NaOH required to move the 20 mL coffee extract from its initial pH value to a pH of 8, as it was suggested by Rao and Fuller (2018) [13].

#### 3.4.4. Phenolic Content

Total phenols were determined by the Folin-Ciocalteu method and expressed as gallic acid equivalents as described by Meireles et al. (2015) [43]

#### 3.4.5. Caffeine Concentration

The caffeine content was determined by spectrophotometer (UNICAM, Helios γ, Boston, MA, USA) at 274 nm. The method used was proposed by Yao et al. (2006) and Koturevic et al. (2017), in which lead acetate solution was used [44,45].

For the construction of the standard curve, the following solutions of known caffeine concentrations were prepared with the corresponding dilutions of the initial solution of 1000 ppm: 0, 80, 160, 240, 320 and 400 ppm.

#### 3.4.6. Color Measurement

The color measurement of the coffee samples prepared during the two series of experiments was performed with a spectrophotometer (ColorLite, sph870, Katlenburg-Lindau, Germany) with the CIELAB color measurement system. Specifically, color is expressed in three values: L* for brightness and a* and b* for the four unique colors of human vision: green–red, blue–yellow respectively. Also, ΔΕ expresses the difference between the color of the sample and a stored standard in the spectrophotometer and is calculated from the sum of the differences of the three values L*, a* and b*. The standard was chosen to be a commercial cold brew coffee. In the section of results and discussion (Section 2) only the values of ΔΕ are presented. Individual L*, a* and b* values are presented in Appendix A.

### 3.5. Sensory Test

Descriptive sensory analysis was performed to determine the optimum extraction parameters, i.e., coffee to water ratio, water total hardness and grinding size, as well as to correlate the operating variables of pressure and vacuum cycles to the sensory attributes of the extract. All analyses were carried out by 6 qualified testers, members of SCAA (Specialty Coffee Association of America). Due to their extensive experience in sensory analysis and cupping, no training of the testers preceded.

Sample preparation: Each sample of coffee extract was prepared three times. The coffee used was ground within 24 h after roasting and all experiments were performed on the same day. After the extraction and before the testing, the samples were placed in the refrigerator to be as close as possible to the form of cold extraction coffee that a consumer would try.

Sample evaluation procedure: All samples were coded randomly using 2 uppercase latin characters and a number. Afterwards, they were evaluated for fragrance, flavor, acidity, aftertaste, body and balance according to the Cold Brew Cupping Protocols [18,46]. Each taster completed a special form of SCAA (Specialty Coffee Association of America) with a descriptive scale of 6–10 with 0.5 secondary scaling and the possibility of adding individual comments regarding specific characteristics of the sample (e.g., indifferent, sweet, strong aroma, fruity, etc.).

### 3.6. Statistical Analysis

ANOVA (one way analysis of variance) with Tukey’s test to compare means was performed on data. Significance was reported at the *p* < 0.05 level. Data are presented as mean values ± standard deviation (SD) obtained from three independent analyses (*n* = 3). Minitab^®^21 (Minitab, Ltd., Coventry, UK) statistical software was used for the statistical analysis. Furthermore, Principal Component Analysis (PCA) was performed. In the present study PCA was applied so as to illustrate the relationship between physicochemical and sensory properties.

## 4. Conclusions

This research reports for the first time the successful parametrization of vacuum assisted cold extraction of coffee to produce a sensorially optimized beverage. Coffee blend, degree of grinding, total water hardness and coffee to water ratio were optimized as the main extraction parameters. The extraction by applying vacuum cycles tested in this study was proved to be a very efficient treatment not only for the acceleration of the traditional and time-consuming method of cold extraction but also for producing a savor beverage comparable to commercial cold brew coffees.

The operating parameters that significantly affect the vacuum assisted extraction were the pressure and the number of vacuum cycles. Low pressure conditions and increased number of vacuum cycles enhanced the extraction process and the transfer of coffee beans’ components to the water (e.g., phenols, caffeine). Principal component analysis successfully correlated sensory profile to operating conditions of the cold extraction process. The coffee beverage produced by applying two vacuum cycles at constant pressure of 205 mbar was distinguished by the specialists panel for the intense sensory profile. Besides the sensory properties, the optimized extraction conditions significantly reduced the brewing time (10 min vs. at least 10 h of traditional cold brewing).

Future research should be conducted to study other factors, such as the effect of roasting degree of coffee beans in the innovative extraction process. The feasibility of using this new process in commercial coffee shop chains and domestically in households should be further investigated. For industrial production a techno-economic analysis is considered mandatory, in order to assess in detail, the economic performance of the suggested brewing technology. Finally, microbial stability and determination of shelf life should be investigated to give an insight in safety issues concerning the consumption of cold brew coffee.

## Figures and Tables

**Figure 1 molecules-27-02971-f001:**
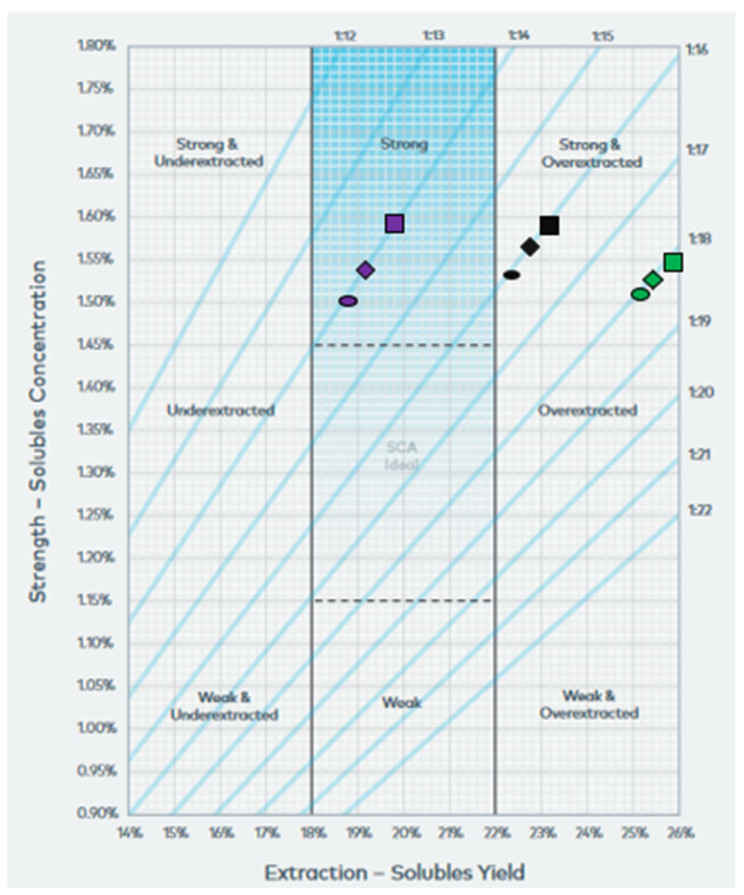
Coffee brewing control chart (for water hardness 70 ppm (■), 109 ppm (⬬) and 290 ppm (◆) [17].

**Figure 2 molecules-27-02971-f002:**
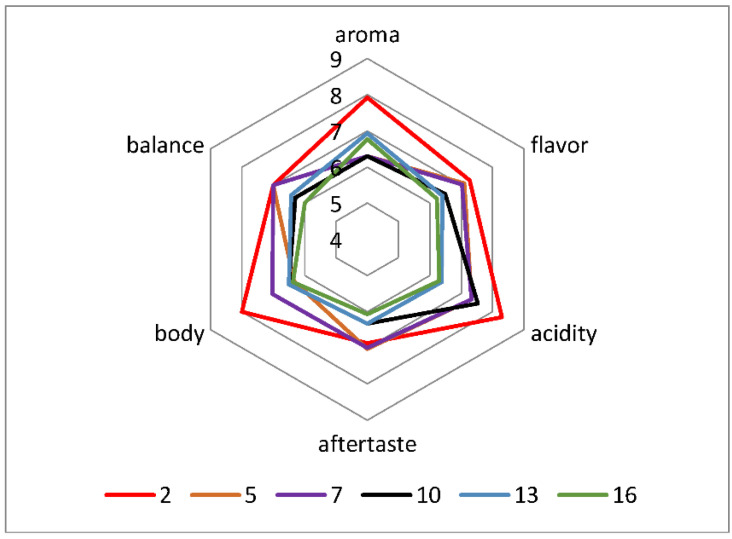
Sensory analysis of extraction under reduced pressure (variation of vacuum cycles).

**Figure 3 molecules-27-02971-f003:**
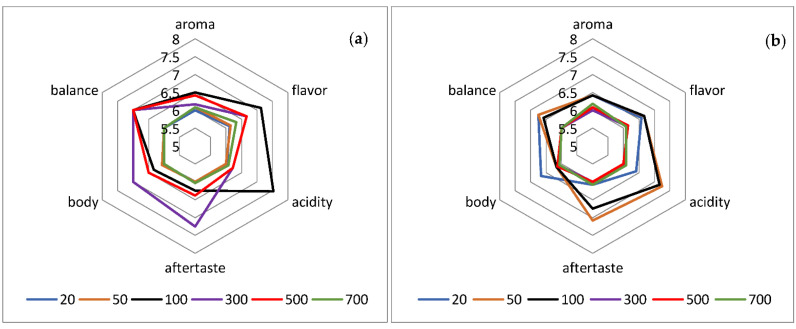
Sensory analysis of extraction under reduced pressure (variation of pressure) (**a**) for 2 vacuum cycles; (**b**) for 7 vacuum cycles.

**Figure 4 molecules-27-02971-f004:**
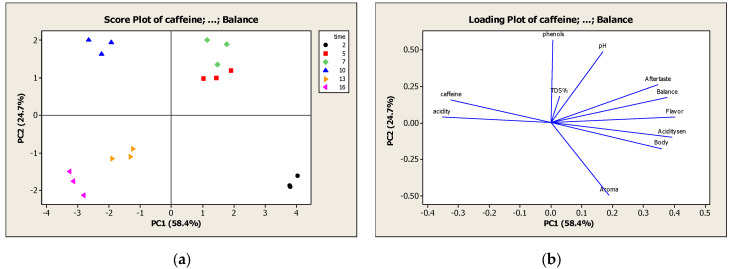
(**a**) Score plot of physicochemical and sensory properties in experiments with variation of vacuum cycles; (**b**) Loading plot of physicochemical and sensory properties in experiments with variation of vacuum cycles.

**Figure 5 molecules-27-02971-f005:**
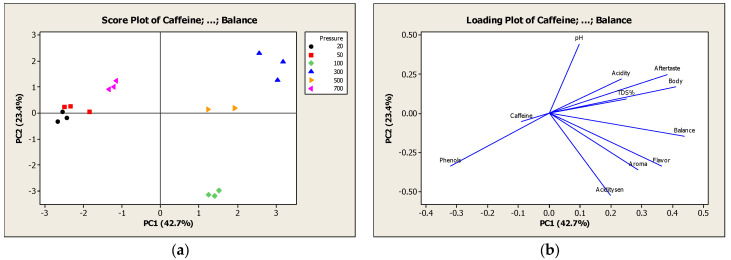
(**a**) Score plot of physicochemical and sensory properties in experiments with variation of pressure; (**b**) Loading plot of physicochemical and sensory properties in experiments with variation of pressure.

**Figure 6 molecules-27-02971-f006:**
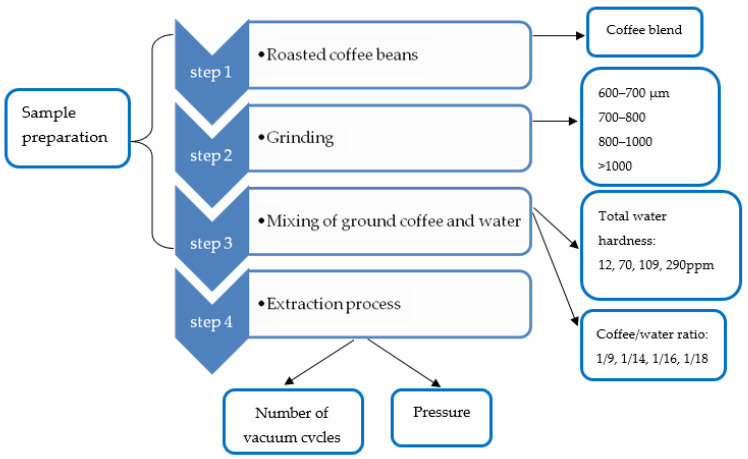
Process Flow Diagram of experimental procedure and the variables to be optimized.

**Table 1 molecules-27-02971-t001:** Tests conditions for the determination of extraction parameters.

First Test: Vacuum cycles selection for initial experiments (300 s each cycle, *p* = 205 mbar)	13 cycles and coffee/water ratio 1/9 7 cycles and coffee/water ratio 1/9
Second Test: Coffee-water ratio and total water hardness, with 7 cycles (300 s each), *p* = 205 mbar	
Blank	1/9
Water hardness: 70 ppm	1/9 1/14 1/16 1/18
Water hardness: 109 ppm	1/9 1/14 1/16 1/18
Water hardness: 290 ppm	1/9 1/14 1/16 1/18
Third Test: Selection of grind size, with coffee-water ratio: 1/14, 7 cycles (300 s each), *p* = 205 mbar	
Water hardness: 109 ppm	600–700 μm 700–800 μm 800–1000 μm >1000 μm

**Table 2 molecules-27-02971-t002:** Optimal extraction parameters ^1^.

Coffee Blend	Coffee/Water Ratio	Total Water Hardness (ppm)	Coffee Grinding (μm)
Columbia-Tres Lomas Estate SHG	1/14	109	>1000

^1^ Extraction operating parameters: vacuum cycles (7), vacuum time (300 s/cycle) and pressure (205 mbar).

**Table 3 molecules-27-02971-t003:** Effect of vacuum cycles on physicochemical properties (avg ± SD, *n* = 3) ^1,2^.

Vacuum Cycles	Caffeine (mg/g Coffee)	Phenols (mg Gallic Acid/g Coffee)	Acidity (mL NaOH)	pH	TDS%	ΔΕ
2	26.7 ± 1.3 ^C^	23.4 ± 0.7 ^C^	2.31 ± 0.06 ^E^	5.02 ± 0.01 ^B^	1.32 ± 0.04 ^B^	21.8 ± 0.6 ^AΒ^
5	29.7 ± 1.1 ^BC^	27.9 ± 0.7 ^AB^	2.52 ± 0.05 ^D^	5.02 ± 0.01 ^B^	1.40 ± 0.08 ^B^	21.5 ± 0.6 ^Β^
7	28.1 ± 1.8 ^C^	30.7 ± 1.8 ^A^	2.89 ± 0.08 ^BC^	5.04 ± 0.01 ^B^	1.51 ± 0.04 ^B^	21.8 ± 1.8 ^AΒ^
10	33.6 ± 0.8 ^A^	31.5 ± 1.7 ^A^	3.03 ± 0.05 ^B^	5.04 ± 0.00 ^A^	1.90 ± 0.21 ^A^	13.9 ± 0.3 ^C^
13	32.0 ± 1.0 ^AB^	24.0 ± 0.8 ^BC^	2.80 ± 0.05 ^C^	5.00 ± 0.01 ^B^	1.49 ± 0.04 ^B^	24.6 ± 0.8 ^A^
16	30.1 ± 0.7 ^AB^	23.0 ± 2.1 ^C^	3.38 ± 0.05 ^A^	4.99 ± 0.01 ^AB^	1.60 ± 0.08 ^AB^	14.4 ± 0.7 ^C^

^1^ Extraction operating parameters: pressure (205 mbar), vacuum time (300 s/cycle), coffee/water ratio (1/14) and water hardness (109 ppm). ^2^ Different superscript letters (A,B, etc.) correspond to significant differences, *p* < 0.05.

**Table 4 molecules-27-02971-t004:** Effect of pressure on physicochemical properties for 2 vacuum cycles (avg ± SD, *n* = 3) ^1,2^.

2 Cycles						
Pressure (mbar)	Caffeine (mg/g Coffee)	Phenols (mg Gallic Acid/g Coffee)	Acidity (mL NaOH)	pH	TDS%	ΔΕ
20	29.0 ± 1.4 ^A^	26.1 ± 0.4 ^A^	2.30 ± 0.08 ^B^	5.09 ± 0.01 ^B^	1.20 ± 0.04 ^A^	19.3 ± 0.6 ^C^
50	24.7 ± 2.2 ^A^	25.9 ± 0.5 ^A^	2.21 ± 0.09 ^B^	5.12 ± 0.01 ^AB^	1.32 ± 0.04 ^A^	16.5 ± 1.1 ^D^
100	25.7 ± 2.8 ^A^	25.0 ± 1.0 ^A^	2.22 ± 0.05 ^B^	5.09 ± 0.01 ^B^	1.31 ± 0.04 ^A^	24.5 ± 0.03 ^A^
300	26.2 ± 2.0 ^A^	20.6 ± 0.8 ^B^	2.59 ± 0.05 ^A^	5.14 ± 0.01 ^A^	1.40 ± 0.04 ^A^	22.1 ± 0.4 ^AB^
500	23.2 ± 2.9 ^A^	20.5 ± 0.4 ^B^	2.20 ± 0.05 ^B^	5.13 ± 0.02 ^A^	1.51 ± 0.22 ^A^	23.7 ± 1.4 ^A^
700	24.7 ± 1.5 ^A^	21.8 ± 0.2 ^B^	2.10 ± 0.05 ^B^	5.15 ± 0.01 ^A^	1.39 ± 0.04 ^A^	20.7 ± 0.7 ^BC^

^1^ Extraction operating parameters: vacuum cycles (2), vacuum time (300 s/cycle), coffee/water ratio (1/14) and water hardness (109 ppm). ^2^ Different superscript letters (A,B, etc.) correspond to significant differences, *p* < 0.05.

**Table 5 molecules-27-02971-t005:** Effect of pressure on physicochemical properties for 7 vacuum cycles (avg ± SD, *n* = 3) ^1,2^.

7 Cycles						
Pressure (mbar)	Caffeine (mg/g Coffee)	Phenols (mg Gallic Acid/g Coffee)	Acidity (mL NaOH)	pH	TDS%	ΔΕ
20	27.7 ± 1.0 ^A^	28.0 ± 0.6 ^C^	2.30 ± 0.02 ^C^	5.02 ± 0.01 ^B^	1.51 ± 0.08 ^AB^	16.1 ± 0.5 ^A^
50	29.1 ± 0.9 ^A^	31.2 ± 0.4 ^AB^	2.49 ± 0.05 ^BC^	5.06 ± 0.01 ^A^	1.51 ± 0.00 ^AB^	14.9 ± 0.1 ^A^
100	28.8 ± 1.2 ^A^	31.4 ± 0.8 ^A^	2.50 ± 0.05 ^BC^	5.06 ± 0.01 ^A^	1.60 ± 0.07 ^A^	17.4 ± 1.3 ^A^
300	28.6 ± 1.2 ^A^	29.1 ± 0.8 ^C^	3.02 ± 0.22 ^A^	5.07 ± 0.01 ^A^	1.32 ± 0.11 ^B^	15.8 ± 1.2 ^A^
500	28.7 ± 1.9 ^A^	29.3 ± 0.6 ^BC^	2.89 ± 0.05 ^AB^	5.06 ± 0.01 ^A^	1.59 ± 0.04 ^A^	16.9 ± 1.0 ^A^
700	25.6 ± 0.4 ^A^	28.1 ± 0.2 ^C^	2.60 ± 0.14 ^BC^	5.07 ± 0.01 ^A^	1.61 ± 0.07 ^A^	16.8 ± 0.4 ^A^

^1^ Extraction operating parameters: vacuum cycles (7), vacuum time (300 s/cycle), coffee/water ratio (1/14) and water hardness (109 ppm). ^2^ Different superscript letters (A,B, etc.) correspond to significant differences, *p* < 0.05.

**Table 6 molecules-27-02971-t006:** Comparison of the extraction method of coffee and their physicochemical properties (avg ± SD, *n* = 3) *.

Extraction Method	Phenols (mg Gallic Acid/g Coffee)	Caffeine (mg Caffeine/g Coffee)	Acidity (mL NaOH)	pH	TDS%
Hot brewing	27.5 ± 1.1 ^A^	20.5 ± 2.9 ^B^	2.5 ± 0.4 ^A^	4.97 ± 0.08 ^B^	1.3 ± 0.3 ^A^
Cold brewing	23.4 ± 0.7 ^B^	26.7 ± 1.3 ^A^	2.3 ± 0.06 ^A^	5.02 ± 0.01 ^B^	1.3 ± 0.04 ^A^
Commercial cold brewing	24.6 ± 0.2 ^B^	28.9 ± 1.6 ^A^	2.5 ± 0.05 ^A^	5.35 ± 0.01 ^A^	1.7 ± 0.0 ^A^

* Different superscript letters (A,B, etc.) correspond to significant differences, *p* < 0.05.

## Data Availability

Data available upon request to corresponding author.

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
