# Peer review of "Optimization of Sensory Properties of Cold Brew Coffee Produced by Reduced Pressure Cycles and Its Physicochemical Characteristics"

_molecules, 2022, doi:10.3390/molecules27092971_

Round 1

Reviewer 1 Report

Cold brewing is a new brewing method that gained attention in recent years. However, the cold brewing method is a time-costuming process, how to reduce the brewing time and get high caffeine concentration is a challenge. The authors provide a new method for cold brewing, using vacuum cycles for the cold extraction of coffee, and compared with conventional methods. The parameters were investigated, including coffee blend, degree of grinding, coffee to water ratio, and total water hardness, pressure and numbers of vacuum cycles. Results showed pressure and number of vacuum cycles are the key parameters, that can enhance the extraction process and transfer the components to the water, indicating the vacuum-assisted cold extraction is a practical method for cold brewing of coffee. Overall, the present manuscript gives readers a good and continuable research, which can be accepted for publication.

Some minor comments:

  1. ‘materials and methods” should be placed following the introduction.
  2. tables can be revised as three-line tables.
  3. the ‘Score Plot of Caffeine; ···” in figure 5 should be revised.

Reviewer 2 Report

The authors present results on the use of vacuum cycles for the cold extraction of coffee, and compare them to conventional methods, studying the effect of the number of cycles and of the applied pressure (vacuum) on the physicochemical characteristics of the cold brew coffee (total dissolved solids (TDS%), pH, acidity, phenol and caffeine content, and color). The manuscript includes results on sensory evaluation by members of SCAA (Specialty Coffee Association of America) and a Principal Components Analysis regarding physicochemical and organoleptic properties.

  The “Materials and Methods” are adequate and sufficiently described in the text, although they should appear as section 2. (just after the “introduction”) instead of being presented after the “Results and Discussion” section.

  The bibliography the authors refer is current and adequate in number to this experimental study.

  In my opinion, this manuscript is rather detailed and interesting, and sets the ground for further studies, that the authors highlight as “future work” in the end of the “Conclusions” section.

 Please address the following remarks.

Q1 - Would this technology be used for industrial production of instant coffee, or in commercial coffee shop chains? I think you could include a sentence on this somewhere on the manuscript, to put this work into context.

Q2 – I understand that vacuum accelerates the extraction, reducing the time needed to prepare a Cold brewed coffee, but do the authors have any idea of the costs of this process, related to the current Cold Brew and the traditional processes? If so, a sentence on this should be included in the manuscript.

Line 23 - correct: “26.66 ± 1.555 mg/g coffee and 23.36 ± 0.794 mg” to “26.66 ± 1.56 mg/g coffee and 23.36 ± 0.79 mg/gallic acid/g coffee”.

Line 31 – correct: “more than ten million tons” to “more than ten million t”

Line 33 – correct “Its consumprtion” to “Its consumption”

Line 46 – correct “14.97g kg-1” to “14.97g/kg” for consistency. If you prefer to use the way you present it, the other units need to be changed to be consistent throughout the manuscript.

Line 47 – correct “proposed by Zhai et al.,” to “proposed by Zhai et al. (2022),” and do the same throughout the text (e.g. line 123, but there are much more).

Line 119 to 153 – It is important to clarify somewhere in this section (currently numbered 2.1.3) that the “Coffee to Water Ratio”  is by Weight. I found this information in reference 17, but it should be included in the text.

Line 164 – correct “>1000 microns particle size”  to “>1000 µm particle size” and substitute “microns” by “µm” throughout the text and tables.

Line 187 – correct “by applying varying number of cycles (300 sec each).” to “by applying a varying number of cycles (300 s each).” The symbols for every unit are internationally defined; “sec” does not exist.

Line 197 – correct “Andrographis paniculate” to “Andrographis paniculate”.

Line 223 – Table 2 – correct all the values in the columns for Acidity and DE regarding the standard deviation according to the significant figures you have from your experimental measurements, namely: “2.3 ±0.06” is either “2.30(if can you write this number with 3 significant figures)±0.06” or “2.3±0.1”.

The same correction is needed for the column Acidity in Table 3 and for some values presented in table 4.

Line 536 – correct “vaccum” to “vacuum”

Line 606 -  correct “charactrers” to “characters”

Line 651 – correct “Acknowledgments” to “Acknowledgements”

Reviewer 3 Report

Dear Authors

The article penned by Kyroglou et al. is interesting and interesting and represents a very useful contribution to increase of knowledge in this field. The experimental part was planned and carried out correctly. The authors make reference to many relevant references. The results of the study are relevant to industry and consumers.  The only substantial  comments I have are on the methodology and conclusions, but I think the authors can fix them quickly. The manuscript is written in very good English, there are a very few minor English errors (they mainly concern punctuation) that if not corrected will detract from your interesting and important paper. Authors should proof-read their paper before resubmitting process. This is why, I recommend to publish this article in Molecules Journal with minor revision

Some comments to improve the article:

Line 32 - consumption

Some content from methodology and results and discussion are repeated unnecessarily, please make appropriate corrections in the text. I have in mind methodological descriptions in results and discussion chapter, that is what the materials and methods section is for. Please see: lines 161-166, line 254 and so on.

Materials and methods

please indicate the origin of the coffee and all equipment used in the study, based on MDPI standards – (city, country)

Conclusions are not conclusions, in their current form they sound more like a summary of the paper. In this section, please provide specific observations from your research, highlight the essence of it. For example: Which coffee preparation conditions do the authors consider best etc?

Good luck with the corrections:)
